# Field Testing of Gamma-Spectroscopy Method for Soil Water Content Estimation in an Agricultural Field

**DOI:** 10.3390/s24072223

**Published:** 2024-03-30

**Authors:** Sophia M. Becker, Trenton E. Franz, Tanessa C. Morris, Bailey Mullins

**Affiliations:** School of Natural Resources, University of Nebraska-Lincoln, Lincoln, NE 68503, USA; trenton.franz@unl.edu (T.E.F.); tmorris10@huskers.unl.edu (T.C.M.);

**Keywords:** soil water content, gamma-ray spectroscopy, field validation

## Abstract

Gamma-ray spectroscopy (GRS) enables continuous estimation of soil water content (SWC) at the subfield scale with a noninvasive sensor. Hydrological applications, including hyper-resolution land surface models and precision agricultural decision making, could benefit greatly from such SWC information, but a gap exists between established theory and accurate estimation of SWC from GRS in the field. In response, we conducted a robust three-year field validation study at a well-instrumented agricultural site in Nebraska, United States. The study involved 27 gravimetric water content sampling campaigns in maize and soybean and ^40^K specific activity (Bq kg^−1^) measurements from a stationary GRS sensor. Our analysis showed that the current method for biomass water content correction is appropriate for our maize and soybean field but that the ratio of soil mass attenuation to water mass attenuation used in the theoretical equation must be adjusted to satisfactorily describe the field data. We propose a calibration equation with two free parameters: the theoretical ^40^K intensity in dry soil and *a*, which creates an “effective” mass attenuation ratio. Based on statistical analyses of our data set, we recommend calibrating the GRS sensor for SWC estimation using 10 profiles within the footprint and 5 calibration sampling campaigns to achieve a cross-validation root mean square error below 0.035 g g^−1^.

## 1. Introduction

### 1.1. Applications Requiring Local Scale Soil Water Content Monitoring

Soil water content (SWC) plays a vital role in the water and energy balance by partitioning the mass and energy fluxes between the land surface and atmosphere [1,2,3], making it an essential state variable for climate and meteorological monitoring and forecasting [4]. SWC data are needed for numerical watershed and land surface models as well as validation of satellite products [5,6]. At the scale of tens of meters, SWC ground observations such as the gamma-ray soil moisture sensor (gSMS) are particularly needed given the historical challenge of collecting data at this scale [6], the current spatial resolution of numerical models and satellite products, and the critical amount of variability that exists at the subfield scale [7].

Hydrological applications for subfield-scale SWC observations include modeling and forecasting of streamflow, floods, and shallow landslides [8,9,10]. Irrigation management also relies upon local-scale SWC information to support crop production, water conservation, and reduction in nutrient runoff and leaching. Within climate monitoring, examples of SWC use include indicating agricultural drought [11] and SWC as a predictor of wildfire risk associated with drought [8]. Local-scale SWC also finds application in validating land surface models and satellite SWC products. This task proves challenging when relying solely on point sensor data because it tends to capture an excessive amount of SWC variability [12,13,14]. While point sensors provide SWC information with too high variability, SWC with resolutions > 1 km do not capture enough variability for accurate description of hydrological, climatic, and meteorological processes impacting many SWC applications, especially across variable landscapes [7,15]. In addition to hydrological, climatic, and meteorological processes, local-scale SWC variability or “hotspots” affect ecosystems (biodiversity and species distribution), plant nutrient cycling and photosynthesis rates [7,16,17]. The SWC monitoring community requires a reliable SWC method that captures critical variability at the subfield scale, supports increased resolution of land surface models, and fills the historic gap in local-scale SWC measurements.

### 1.2. Existing Soil Water Content Sensors

Numerous in situ and remote SWC sensors are currently available, and we refer the reader to the literature for more in-depth analysis [18,19,20,21,22]. Advantages, disadvantages, and accuracy of common SWC sensors are briefly presented here for comparison with the gSMS (Table 1). Recent reviews of soil moisture sensing technologies have found that in situ sensing is dominated by dielectric methods, including time domain reflectometry (TDR), frequency domain reflectometry, and capacitance probes [8,22,23,24]. The authors of [25] found that accuracy for dielectric methods generally ranges from ±0.01 to ±0.19 m^3^ m^−3^ and improves to between ±0.01 and ±0.04 m^3^ m^−3^ for studies with field-specific calibrations. Similarly, ref. [26] showed that, with site-specific calibration, portable dielectric probes perform with a root mean square error (RMSE) of 0.025 m^3^ m^−3^ in a sandy loam soil. In addition to dielectric methods, point-scale SWC can be estimated with neutron scattering, active gamma-ray attenuation, tensiometer, resistance block, and thermal dissipation block methods [19,22,27]. Additional in situ soil moisture sensors in the early stages of development include actively heated fiber optics, hydrogels, high-capacity tensiometers, radio frequency identification, and novel pairings of acoustic, radio, and seismic transceivers [23]. Lack of spatial representativeness by point SWC sensors and the necessity of measurements that integrate large areas (>100 m^2^) to fill the gap between the point and remote sensing scale are identified throughout recent reviews [28,29,30,31]. Methods for capturing SWC above the point scale include wireless sensor networks, on-the-go dielectric methods, electromagnetic induction, ground penetrating radar (GPR), global positioning system (GPS), and cosmic-ray neutron sensing [19,21,32,33,34]. Given the accuracy of existing in situ and geophysical methods (Table 1), emerging SWC sensors such as the gSMS should aim for RMSE ≤ 0.04 m^3^ m^−3^ to provide hydrological applications with useful subfield-scale data.

### 1.3. Spatial Scale of Soil Water Content Monitoring

Local SWC spatial variability results from soil heterogeneity, topography, landcover, meteorological forcing, and temporal dynamics of mean SWC [12]. These factors produce an SWC spatial correlation length (defined by the range of the semi-variogram) that is site-specific and dynamic. The site-specificity of SWC spatial correlation length is demonstrated in one study sampling 60 different sites in North America, where SWC spatial correlation length ranged from 2 to 78 m [43]. Spatial variability is highest in the intermediate SWC range and lower at SWC extremes [44], so dynamics of mean SWC due to factors such as vegetation impact SWC variability. To capture site-specific and dynamic SWC spatial variability, specific monitoring scales and extents are necessary [45].

The available SWC monitoring methods can be arranged on a continuum comprised of the point, subfield, field (100 s of m), and coarse (>1 km) scales. Point-scale measurements (TDR and other dielectric sensors with a volley-ball-sized sensing volume) [18] and coarse measurements are currently more prolific than methods characterizing the local (subfield and field) scale. Coarse resolution SWC is available from satellite missions using microwaves, such as Soil Moisture Active Passive (SMAP), which obtains SWC estimates with ~10 km resolution for the top 5 cm of the soil [46,47]. Vergopolan et al. [7] utilized SMAP-HydroBlocks, a 30 m resolution SWC data set for the top 5 cm of the soil across the United States [48] and found that, on average, 48% of SWC spatial variability information is lost when moving from the 30 m to the 1 km scale. Considering emerging 30 m resolution land surface models and the critical SWC variability seen at the 30 m scale, obtaining SWC observations at this scale is crucial.

Potential SWC monitoring methods at the local 30 m scale include GPS, the cosmic-ray neutron sensor (CRNS), and GRS. GPS covers a horizontal area with a radius of 50 m for antenna height of one meter and a sensing depth that varies from a few millimeters to 7 cm depending on water content [23]. CRNS estimates SWC for a circumference with a radius around 200 m depending on installation height and a moisture-dependent sensing depth of 12 to 70 cm [49] or 15 to 83 cm [50]. The CRNS is a well-established and accurate method and senses a deeper signal than GPS [33], but challenges arise for fields that are smaller than the CRNS footprint, especially when irrigation is occurring or when SWC is changing in distant areas of the footprint [51,52]. Fortunately, the gSMS is equipped to describe the subfield scale at greater sensing depth than GPS and match the 30 m scale more closely than CRNS. When installed at a height ~2 m above the soil surface, the gSMS has a circular footprint radius of 24 m and maximum sensing depth of 70 cm [53]. The gSMS has been deployed within in situ, roving, and unmanned aerial vehicle campaigns and demonstrates potential to accurately measure subfield SWC [53,54,55]. Still, very few attempts have been made to quantify the relationship between SWC and gamma-ray intensity [28]. Before gSMS subfield SWC is adopted for capturing the critical variability required for hydrological applications, the underlying theory for estimating SWC from GRS must be further validated with field experiments.

### 1.4. Theoretical Relationship between Detected Gamma Radiation and Soil Water Content

Theoretical gamma-ray intensity detected by the gSMS is determined by the cross-sectional area of the detector, detector efficiency, volume from which gamma rays originate, height of the detector, and attenuation of the gamma rays in air and in soil [56]. The attenuation of gamma rays in soil is strongly influenced by the SWC because the electron density of water is 1.11 times that of a typical dry soil [57]. This relationship between SWC and gamma-ray intensity is expressed in terms of the change in detected gamma-ray intensity due to change in hydrogen content within the sensing volume. Examples of this expression for a point source can be found in [57,58].

The general equation for total gamma-ray intensity (s^−1^) detected from a radioactive volume with a flat surface, $I_{total}$, from a height, h, above the ground is [59,60]:(1)$$I_{total}=\;\frac{A\epsilon \gamma }{4\pi }\int_{R_{min}}^{R_{max}}\int_{0}^{\theta_{1}}\int_{0}^{2\pi }\frac{1}{R^{2}}e^{-\mu_{a}\rho_{a}r_{a}}e^{-\mu_{g}\rho_{g}r_{g}}\sin\left(\theta \right)R^{2}d\varphi d\theta dR,\;\;\;\;\;\;\;\;\;\;\;$$
with $R_{min}\geq \;h/cos\theta \;$, $r_{a}=\;h/cos\theta \;$, and $r_{g}=R-h/cos\theta \;$. A is the cross-sectional area of the detector (m^2^), ϵ is the efficiency of the detector for a given energy, γ is the number of gamma rays emitted per cubic meter of source material per second, R is the total distance between the detector and gamma-ray origin (m) and equal to the sum of $r_{a}$ and $r_{g}$, θ is the angle of the detector with respect to the normal of the soil surface (radians), and φ is the third spherical co-ordinate. The energy-dependent mass attenuation coefficients (m^2^ kg^−1^) of the air and ground are $\mu_{a}$ and $\mu_{g}$, respectively; $\rho_{a}$ and $\rho_{g}$ are the densities (kg m^−3^) of the air and ground, respectively; and $r_{a}$ and $r_{g}$ are the distances (m) travelled by gamma rays in the air and ground, respectively. Limits of *R* and θ can be chosen to model a constant source depth [59], an infinite source depth [61], or a depth defined by an isoline along which the total gamma-ray attenuation by the ground and air is constant [53]. The isoline approach enables estimation of the source volume and corresponding footprint characteristics.

### 1.5. Additional Factors Affecting Gamma Radiation Estimation of SWC

Current research on GRS sensing of SWC is disentangling complicating aspects such as the height-dependent sensing volume, SWC and soil heterogeneity, and sources of signal attenuation besides pore soil water (e.g., biomass). Historic studies regarding GRS sensing of SWC were airborne [58,62], except for [57], who collected data with a proximal portable gamma-ray spectrometer and compared SWC estimates at the 0.1 and 0.25 m soil depths. Loijens [57] also explored vertical soil heterogeneity numerically; error in estimated gravimetric water content using total gamma-ray intensity due to violation of the homogenous SWC profile assumption was concluded to be around 1.8% typically and 4 to 5% in extreme cases. From there, we jump to the present day, where [53] recently addressed the need to better quantify the gamma-ray sensing volume and make height corrections for detector heights between 0 and 40 m. Recent research has also addressed the problem of biomass correction with a Monte Carlo simulation approach [63]. By modeling biomass as an equivalent water layer above the soil surface, ref. [63] showed that neglecting to correct for biomass leads to an over 30% positive systematic bias in SWC estimation. The authors of [63] write the equation for gravimetric SWC from ^40^K intensity at measurement time, $w_{\gamma K}^{\Lambda }\left(t\right)$, with the Monte-Carlo-derived biomass water correction factor, $\Lambda_{K}\left(BWE\left(t\right)\right)$, as:(2)$$w_{\gamma K}^{\Lambda }\left(t\right)=\frac{S_{K}^{Cal}\cdot \Lambda_{K}\left(BWE\left(t\right)\right)}{S_{k}\left(t\right)}\cdot \left[\Omega +w_{G}^{Cal}\right]-\Omega ,$$
where:(3)$$\Lambda_{K}\left(BWE\left(t\right)\right)=\left(-0.0120\pm 0.0001\right)*BWE+1.0$$
and BWE is biomass water equivalence in mm at measurement time. $S_{K}^{Cal}$ is the ^40^K net count rate (cps) at calibration time, $w_{G}^{Cal}$ is the gravimetric water content at calibration time (g g^−1^), and $S_{k}\left(t\right)$ is the ^40^K net count rate at measurement time (cps). In addition to pore water, lattice (structural) water and soil organic carbon water are also attenuating hydrogen pools [64]. The ratio of the mass attenuation coefficient of soil to that of water, Ψ, and the fraction of structural water, $f_{H_{2}O}^{struct}$, are incorporated in the dimensionless Ω term used in Equation (2).
(4)$$\Omega =\Psi +\left[1-\Psi \right]\cdot f_{H_{2}O}^{struct}.$$

In application, we assume that Ψ is equal to that of pure silica soil. Applying Equation (2) to four observations in a tomato field, a 4% average relative discrepancy was found between GRS-estimated SWC and gravimetric SWC [63].

In contrast to biomass water content and detector height, factors such as soil mineralogical composition and fluctuation in atmospheric water vapor and density have been shown to have negligible influence on the SWC estimate at heights below 40 m [60]. Another variable in the gSMS method is establishing standard data analysis practice for calibration. Current practice is to use Full-Spectrum Analysis (FSA) to determine specific activity of individual radionuclides because higher count rates are obtained as opposed to the values from the traditionally used Windows Analysis [65,66]. Up to this point, studies have not considered which analysis method is best suited for SWC estimation. The essential task of validating theory and establishing a reliable method for SWC estimation using GRS remains unfinished, in part due to the various factors that must be accounted for in the calibration equation and overall methodology.

### 1.6. Approach

Research on SWC sensing with the CRNS has shown through both field-based and modeling studies the importance of correcting for a variety of factors such as biomass water content, soil organic carbon water, lattice water, and effective sensing depth controlled by soil water content [64,67,68]. The calibration equation for the gSMS must incorporate similar correction to be robust to field conditions, including those where biomass water content may be changing rapidly and where soil bulk density may change over time due to field operations. In addition to focused SWC monitoring with GRS, correction algorithms for SWC are a pressing need in environmental gamma-ray spectroscopy research as a whole [69]. Considering ease of access to in situ, portable, and UAV-mounted gamma-ray spectrometers and potential of Monte Carlo simulations for understanding gamma-ray transport, the GRS community is poised to make significant progress in establishing an accurate subfield SWC sensing method built upon a validated theoretical model and incorporated correction factors.

Within the context of the need for gSMS local-scale SWC observations and the remaining questions for the gSMS method implementation, we present a field validation study in eastern Nebraska, United States. Gravimetric SWC data over a three-year period that include soybean and maize growing seasons are compared with ^40^K specific activity (determined with FSA) from an in situ gSMS and a theoretical calibration function similar to [63]. We hypothesize that our robust calibration data set over a range of SWC and vegetative conditions will allow us to validate and/or improve the calibration function with biomass correction. We also utilize our data set with various statistical analyses to offer insight into practical calibration methods and best practice.

## 2. Materials and Methods

### 2.1. Site Description

In this study, we tested the gSMS at an agricultural field within the Eastern Nebraska Research and Extension Center near Mead, Nebraska, United States. The US-Ne3 site is within the United States Department of Agriculture Long-term Agroecosystem Research (LTAR) Network as well as the Ameriflux network. US-Ne3 is a no-till, rainfed site with a maize–soybean rotation. Length of the frost-free growing season is approximately 161 days and the majority of precipitation falls during the April to September growing season [70]. The mean annual precipitation is 760 mm and the mean annual temperature is 10 °C. The highest average monthly temperature (31 °C) occurs in July and the lowest (−9 °C) occurs in February [70]. The soils are mollisol silt loams and silty clay loams (average texture of 30.1% clay, 58.7% silt, and 11.2% sand for 0–30 cm depth).

### 2.2. Instrumentation

The gamma-ray sensor used in this study is the 100 mL cesium iodide (CsI) gSMS from Medusa Radiometrics. The gSMS was installed at a height of 1.86 m at the site in July 2021 (Figure 1), where it sums gamma-ray spectra over 15-min intervals and sends the data wirelessly to the online Medusa portal every hour. The Medusa gSMS provides real-time data processing, including energy stabilization and non-negative least squares FSA to calculate specific activity (Bq kg^−1^) of ^40^K, ^238^U, and ^232^Th. Here, we use only ^40^K in SWC estimation to avoid the error introduced by radon into the measured ^238^U concentration and to streamline the initial field validation. For reference, 1% mass concentration of potassium in rock converts to ^40^K specific activity of 313 Bq kg^−1^ [66]. The calibration file including the standard spectra required for processing was provided by Medusa with purchase of the gSMS and is recommended to be reliable for five years. Based upon installation height and [53], the horizontal footprint of the gSMS for dry soil is estimated to be a circle with a radius of 24 m and the vertical sensing depth is approximately 35 cm.

The long-term agricultural research data collection at the site includes time-domain reflectometry, a CRNS, an eddy-covariance tower, and biomass sampling. Six intensive measurement zones (IMZ) are established at the site, from within which destructive biomass sampling is conducted every 10 days during the growing season.

### 2.3. Gravimetric Water Content Sampling

Gravimetric water content was measured with a sample pattern designed to characterize the estimated gSMS sensing volume in the horizontal and vertical directions, assuming 95% of the gamma-ray signal originates within 24 m horizontally and 35 cm vertically. Samples were collected every 60° at radii 0, 2, 5, and 12 m away from the gSMS for a total of 133 samples for each campaign (Figure 2). In the vertical direction, seven total samples were collected every 5 cm down to a maximum depth of 35 cm below the surface. Sample tins were then returned to the lab, weighed, and dried at 105 °C for 48 h to determine the ratio of mass of water ($M_{w}$) to mass of dry soil ($M_{s}$) in g g^−1^. Gravimetric water content, $\theta_{g}$, is defined as:(5)$$\theta_{g}=\frac{M_{w}}{M_{s}}.$$

The $\theta_{g}$ for a given sample date was found by calculating a depth-weighted mean for each vertical SWC profile and then taking the arithmetic average of the 19 sample locations. The depth-weighting method is described in Section 2.5.

### 2.4. Biomass Characterization

We conducted 27 sampling campaigns between 2021 and 2023 in a variety of vegetative conditions. Vegetation conditions for the sampling days are described in Table 2. Maize stover describes conditions between maize harvest and soybean planting due to maize stalks left in the field. Bare soil conditions occurred after soybean harvest. Fresh and dry aboveground biomass data from destructive sampling were used to calculate biomass water equivalence (BWE) in mm following [71]. The BWE data points were linearly interpolated to estimate the BWE at time of soil sampling. For maize stover, we estimate that a small amount of BWE remains in the plant water and cellulose and decreases as the stalks continue to dry through time, so BWE was assumed to be 0.5 mm in maize stover immediately following maize harvest and to linearly decrease to a value of 0.01 mm at the time of soybean harvest. The total BWE during the soybean growing season is the sum of the estimated maize stover BWE and the soybean BWE. Bare soil conditions were assumed to be equivalent to 0 mm BWE. Residue was not quantified in this study.

### 2.5. Depth-Weighting of Soil Samples

Gravimetric samples were depth-weighted to reflect the fact that the soil volume closer to the detector contributes more of the detected gamma signal than soil farther away from the sensor. However, we do not incorporate the change in effective sensing depth due to changes in SWC. An example of depth-weighting with the added complexity of variable effective sensing depth is found in [68]. The total gamma flux density (s^−1^) at height h (m), $\Phi_{tot}\left(h\right),$ for a semi-infinite homogenous source volume is [72]:(6)$$\Phi_{tot}(h)=\frac{A_{v}P_{\gamma }}{2\mu_{s}}\int_{0}^{\frac{\pi }{2}}sin\theta e^{\frac{-\mu_{a}h}{cos\theta }}d\theta .$$

The energy-dependent linear attenuation coefficients for air and soil are $\mu_{a}$ and $\mu_{s}$, respectively ($m^{-1}$), and the flux is calculated for detector angle θ from 0 to $\frac{\pi }{2}$, where θ is the angle measured from the vertical line normal to the soil surface. $A_{v}$ is the unit volume activity ($Bqm^{-3}$) and $P_{\gamma }$ is the gamma-ray intensity in “number of emitted gamma-rays/Bq”. For simplicity, we exclude horizontal variation in contribution to gamma-ray flux and focus on the vertical direction. To examine the vertical direction, we find the gamma flux, $\Phi \left(h,t\right),$ for a source with fixed depth and infinite radius for a detector placed at height h produced within a soil thickness t (m) in simplified notation for the implicit gamma energy dependence [54,72]:(7)$$\Phi \left(h,t\right)=\frac{A_{v}P_{\gamma }}{2\mu_{s}}\int_{0}^{\frac{\pi }{2}}sin\theta e^{\frac{-\mu_{a}h}{cos\theta }}\left[1-e^{\frac{-\mu_{s}t}{cos\theta }}\right]d\theta .$$

Then, the cumulative percent contribution to the total detected gamma flux at a given depth, $\Phi_{rel}\left(h,t\right)$, is the gamma flux in overlying soil thickness, t, over the total gamma flux:(8)$$\Phi_{rel}\left(h,t\right)=\frac{\Phi \left(h,t\right)}{\Phi_{tot}(h)}.$$

Let n be the number of sampling intervals and let $0\leq l_{0}<\ldots l_{n}$ be an arbitrary sampling partition of the interval $\left[0,t_{max}\right]$. For instance, our sampling partitions are (0, 5, 10, …, 30, $t_{max}$). $t_{max}$ represents the soil thickness from which over 99% of the gamma signal originates, which is calculated for our study to be 0.33 m for h = 1.86 m, bulk density 1.37 g cm^3^, and the average water content observed. To understand the extent to which our sensor’s effective sensing depth may vary, the theoretical depth from which different percentages of the total detected gamma flux originates was calculated for different bulk densities and total water contents with Equation (8) for our detector height (Figure 3).

Then, for sample i=0,1, 2, …, n−1, the area under the cumulative contribution percentage curve from the corresponding sampling partition is:(9)$$A_{i}=\int_{l_{i}}^{l_{i+1}}\Phi_{rel}\left(h,t\right)dt$$

To calculate the weight for a given sample, we first find the rectangular area that represents a weight of one for a given interval, which is the product of the partition width and the height, *b*, set equal to 1 (100% contribution to the detected gamma flux):(10)$$B_{i}=\left(l_{i+1}-l_{i}\right)b$$

The constant, a, that conserves the weights is:(11)$$a=\frac{1}{1-\int_{0}^{t_{max}}\Phi_{rel}\left(h,t\right)dt}$$
with the upper limit of t equal to $t_{max}.\;$ The weight, w, for a given sample, i, is then:(12)$$w=\;a\left(B_{\;i}-\;A_{\;i}\right)$$

The calculated weights for the gravimetric soil samples are given in Table 3, where $\mu_{a}=\;$0.0064 m^−1^ and $\mu_{s}=\;$9.34 m^−1^. The depth-weighted average water content, $\theta_{wt}$, for a vertical profile with  j =1, 2, 3, …, n for n sampling intervals is calculated as:(13)$$\theta_{wt}=\sum_{j=1}^{n}w_{j}\theta_{j},$$
where $w_{j}$ is the weight for sampling interval *j* (Equation (12)) and $\theta_{j}$ is the gravimetric water content value for sampling interval *j*. Note, for a more complete treatment of support volume, Monte Carlo particle transport codes like MCNP or URANOS may be used [50,54,63,73].

### 2.6. Auxiliary Data

Estimation of SWC requires additional site-specific information, including bulk density, soil lattice water content, and soil organic carbon water. This study uses the values in Table 4 from chemistry and bulk density analyses conducted at the site in 2023. Bulk density was sampled three times in 2023 at the beginning, middle, and end of the growing season using a core sampler with 2 in diameter and six depth intervals of 5 inches each at six locations within the gSMS footprint. The bulk density profiles were depth-weighted and averaged, and then the three bulk density samples were averaged to represent the mean field conditions throughout the study (Table 5). A soil chemistry sample was collected by aggregating soil from each profile location (total 100 gm) on 1 sampling date (15 May 2023). The sample was analyzed for lattice water and SOC at Actlabs (Ontario Canada) following the same methods used for CRNS [64].

### 2.7. Calibration Equation

The calibration equation evaluated in this study is functionally identical to Equation (2) from [63] and can be derived by assuming that $w_{G}^{Cal}$ is equal to zero and the value used for $S_{K}^{Cal}$ is the count rate for ^40^K in dry soil. The notation has been changed in an attempt at simplification and better alignment with [53,59]. The calibration equation without the $f\left(BWE\right)$ correction factor can also be derived by following Appendix C from [60], based on the integration along an isoline of Equation (1). Note that the equation used in this study describes total soil water content (instead of only pore water content) similar to CRNS literature [64,74,75]. Total soil water content is defined as:(14)$$\theta_{tot}=\theta_{g}+\theta_{lattice}+\theta_{SOC},$$
where $\theta_{g}$ is the gravimetric pore water content, $\theta_{lattice}$ is the equivalent water in the soil mineral structure, and $\theta_{SOC}$ is the soil organic carbon water equivalent (g g^−1^). The weighted $\theta_{lattice}$ and $\theta_{SOC}$ values from Table 4 were added to the experimental depth-weighted $\theta_{g}$ values to obtain the $\theta_{tot}$ observations for evaluation of the calibration function. The calibration function is written as:(15)$$\theta_{tot}=\left(\frac{I_{0}\cdot f\left(BWE\right)}{I_{t}}-1\right)\frac{{\left(\mu /\rho \right)}_{s}}{{\left(\mu /\rho \right)}_{w}},$$
where $I_{t}$ is the ^40^K specific activity at measurement time (Bq kg^−1^), $I_{0}$ is the ^40^K specific activity in dry soil, and $f\left(BWE\right)=\Lambda_{K}\left(BWE\left(t\right)\right)$ found by [63]. The energy-dependent mass attenuation coefficients of soil and water are ${\left(\mu /\rho \right)}_{s}$ and ${\left(\mu /\rho \right)}_{w}$, respectively (m^2^ kg^−1^). For ${\left(\mu /\rho \right)}_{s}$, the value corresponding to SiO_2_ and the energy of the ^40^K peak, 1.46 MeV, is used (0.005257 m^2^ kg^−1^). The value for ${\left(\mu /\rho \right)}_{w}$ at 1.46 MeV is 0.005836 m^2^ kg^−1^ [76].

### 2.8. Evaluation of Calibration Equation

All data analyses were performed using R Statistical Software v4.0.2 [77]. Equation (15) was fit to the data using $f\left(BWE\right)=\left(-0.0120\pm 0.0001\right)*BWE+1.0$ [63]. To fit parameters, the sum of the absolute value of the residuals was minimized with Shuffled Complex Evolution using the function, “sceua”, within the rtop package (v0.6.6) [78]. The goodness of fit was determined by leave-one-out cross-validation, where the errors of the single “left out” test predictions were used to calculate RMSE and adjusted R^2^ (Adj. R^2^). In addition to goodness-of-fit statistics, the model was evaluated by plotting the residuals and checking for remaining trends. When a statistically significant linear trend was found in the residuals, an additional parameter, a, was added to the calibration equation so that the updated model equation was:(16)$$\theta_{tot}=\left(\frac{I_{0}\cdot f\left(BWE\right)}{I_{t}}-1\right)\frac{{\left(\mu /\rho \right)}_{s}}{{\left(\mu /\rho \right)}_{w}}*a.\;$$

### 2.9. Recommended Sample Sizes for Parameter Calibration

The gravimetric sample data were used to determine the number of point samples required to characterize the gSMS footprint. For each sampling day, the depth-weighted average of the 19 total SWC samples was considered the true SWC. Then, for profile sample sizes 3 to 16, the profile sample was randomly selected 10,000 times without replacement. Each selected profile sample was depth-weighted, averaged, and compared with the true water content. The average RMSE and relative error was then calculated for each profile sample size for each sample day.

The gravimetric sample data were also used to evaluate the number of calibration days required. For calibration sample sizes 3 to 27, the calibration sample was randomly selected 10,000 times without replacement. For each of the calibration days in the sample, 10 profiles were randomly selected and depth-weighted. The SWC values from the calibration days were then used to fit the model. The RMSE was calculated for the fitted model predictions of all 27 SWC observations.

## 3. Results

### 3.1. Comparison of Observations with Calibration Equation

All observed soil moisture profiles from gravimetric sampling are provided in Figure 4a–c, showing that, overall, a variety of heterogeneous soil moisture profiles were captured with wetting fronts visible in some instances (e.g., 10 July 2023 and 24 July 2023). The relationship between observed $\theta_{tot}$ and ^40^K specific activity follows the general behavior expected by Equation (15). However, seven of the samples do not have uncertainty intervals that intersect the 95% confidence interval for Equation (15) with the biomass correction omitted, and the samples deviate from Equation (15) at both the dry and wet ends of the curve (Figure 5).

Inclusion of the biomass correction factor improves the RMSE from 0.052 g g^−1^ to 0.045 g g^−1^ and the Adj. R^2^ from 0.05 to 0.25. The increased estimation error by the first calibration equation (Equation (15) with biomass correction included) at the dry and wet ends of the SWC range is visible in the residuals (Figure 6a), which reveal a significant linear trend with respect to predicted total water content and ^40^K specific activity (*p*-value < 0.01). The average goodness-of-fit statistics and parameter values found from fitting the model with the shuffled complex evolution algorithm and leave-one-out cross-validation are given in Table 6.

When the second calibration equation (Equation (16)) is fit to the data, the trend in the residuals is removed (Figure 6b). The fitted values for $I_{0}$ and *a* are given in Table 6, along with goodness-of-fit statistics that demonstrate improved RMSE and Adj. R^2^ values compared to the first calibration equation.

Predicted $\theta_{tot}$ from the second calibration equation usually describes the observed $\theta_{tot}$ better than the first calibration equation when plotted as a time series (Figure 7). Three samples in 2023 that were collected within four hours of precipitation events were underpredicted by both the first and second calibration equations (Figure 7c).

### 3.2. Sample Size Analyses

The analysis of profile sample size shows that the true SWC in the gSMS footprint can generally be estimated with a relative error less than 3% with only 10 profiles (Figure 8). Calibration sample size analysis shows that RMSE decreases sharply when increasing from three to five calibrations (Figure 9). The 27 calibrations (19 profiles each) can be predicted with an RMSE of 0.038 g g^−1^ using three calibrations and an RMSE of 0.035 g g^−1^ using five calibrations (10 profiles each). The minimum RMSE, with 27 calibrations, is 0.031 g g^−1^.

## 4. Discussion

### 4.1. Describing Field Gamma-Ray Behavior

From a statistical modeling perspective, the ratio, ${\left(\mu /\rho \right)}_{s}/{\left(\mu /\rho \right)}_{w}$, should be modified in some way to eliminate a residual trend in our field data (Figure 6). The mass attenuation adjustment parameter, a, fit with our experimental data tells us that the “effective” ratio ${\left(\mu /\rho \right)}_{s}/{\left(\mu /\rho \right)}_{w}$ is approximately 60% of the ratio ${\left(\mu /\rho \right)}_{s}/{\left(\mu /\rho \right)}_{w}$ corresponding to 1.46 MeV and SiO_2_ material (Table 6). The need to modify the mass attenuation ratio was not due to a difference in soil mineralogical composition compared to a pure SiO_2_ soil; the difference in ${\left(\mu /\rho \right)}_{s}$ for different soil mineralogical composition is less than 1% for energies from 0.3 to 3 MeV [60]. Therefore, we consider other explanations for the reduced effective ${\left(\mu /\rho \right)}_{s}/{\left(\mu /\rho \right)}_{w}$ such as our gamma-ray analysis method. Because FSA considers the entire ^40^K spectrum, the ${\left(\mu /\rho \right)}_{s}/{\left(\mu /\rho \right)}_{w}$ that should be used with ^40^K specific activity values from FSA is potentially some weighted average of the $\left(\mu /\rho \right)$ values across all the energy levels represented in the standard ^40^K spectrum. This concept could be further explored in future work by using only the information from the ^40^K peak instead of the full spectrum.

Regardless of questions surrounding the mass attenuation parameters, our results demonstrate the success of the biomass correction factor from [63] for maize, maize stover, and soybean. The biomass correction factor markedly improved the goodness-of-fit statistics and, when the biomass correction factor is fit to the experimental data along with $I_{0}$ in the first calibration equation, the fitted value is within 0.002 of the 0.012 value given by [63]. Given the high performance of the biomass correction approach, we conclude that modeling vegetation as a layer of water above the soil surface is a sufficient method for removing the influence of vegetation on the gSMS signal in maize and soybean at our site.

### 4.2. Method Limitations

SWC estimation with the gSMS is limited by the need for calibrated parameters ($I_{0}$ and a). The number of fitted parameters in the calibration equation determines the number of calibration sampling campaigns required to fit those parameters. The current suggestion for at least five calibration campaigns limits the method to dedicated research contexts. Similarly, calibrating both the $I_{0}$ and a will pose significant challenges to spatial mapping of SWC with the gSMS. The site-specific nature of our study means that the ${\left(\mu /\rho \right)}_{s}/{\left(\mu /\rho \right)}_{w}$ and biomass water corrections should also be validated in other soil and vegetation conditions. Future work should also investigate if the a parameter is consistent across study sites or correlated to properties such as soil texture or underlying mineralogy.

Handling heterogeneity in the SWC profile and in the contribution of source material at varying distances from the gSMS is also a point of limited understanding. Besides [57] calculating the percent error expected from a heterogeneous soil water profile when assuming a homogeneous profile, little research has explored the potential impact on gamma-ray SWC estimates. An instance of SWC heterogeneity effects was the poor predictability of samples near precipitation events and the slight negative impact they had on our model fit when they were included in the calibration data set vs. when they were removed as outliers. Both the characteristics of increased vertical and temporal variability were likely at play near precipitation events. Similarly, the optimal method for sample weighting is currently open to discussion and could be enhanced by applying the knowledge that contribution to the detected gamma signal is a function of both distance to the gSMS and dynamic overall density (bulk density and SWC) of the material. Achieving this would require Monte Carlo simulation codes such as MCNP or URANOS for CRNS. Another limitation, particularly in relation to precipitation events and utilizing Monte Carlo simulation, is the unknown impact of neglecting the hydrogen pool associated with intercepted water and dew on the canopy.

### 4.3. Favorable Characteristics

The unbiased RMSE value of 0.033 g g^−1^ (0.045 m^3^ m^−3^ for a bulk density of 1.37 g cm^−3^) places the gSMS method in this study among other in situ and remote sensing SWC methods in accuracy (Table 1). Equation (16) is now feasible to use for dedicated research purposes with row crops where the benefit of the gSMS information outweighs the cost of calibration sampling. Because of the SWC estimation accuracy and improved description of gamma-ray behavior in the field using Equation (16) compared to the theory-based Equation (15), this study opens an avenue for measuring SWC at the novel subfield scale and provides concrete motivation for further development of GRS for hydrological applications. Future development is promising because technical challenges of gamma-ray sensing are resolving with the availability of smaller detectors and software for processing gamma-ray spectra. Additionally, the gSMS footprint is fairly well described by theory and Monte Carlo simulations [53,54].

### 4.4. Recommendations for Stationary gSMS Operation

Calibration sampling for gravimetric water content is the crux of obtaining accurate SWC estimates with the gSMS. Our results show that two parameters must be fit at each site, which leads us to recommend five sampling campaigns for an error of 0.032 g g^−1^. Additional sampling campaigns may reduce the unexpected error marginally. Equation (16) best represents our current understanding of the relationship between gamma-ray specific activity and SWC by preserving the known mass attenuation coefficient of SiO_2_ for a collimated beam and a “fudge factor” (i.e., additional parameter, a) to account for an unexplained relationship between the residuals and ^40^K specific activity values from FSA. The sampling size analysis in Section 3.2 provided a conservative value of 10 for the number of locations that should be sampled to measure actual soil water content within the gSMS footprint with a relative error less than 3%. We do not recommend a single or a few point sensors such as a TDR sensor for calibration given the discrepancy in sensing volume between point sensors and the gSMS and the high degree of variability in point data as discussed in Section 1.1 and Section 1.2. At each location in the sampling design, gravimetric water content samples should be collected at five-centimeter intervals. The depth-weighting approach used here is recommended and the weights given in Table 3 can be used; the R code to calculate the weights for specific cases is provided on GitHub (see Appendix A). In addition to gravimetric sampling, BWE must be estimated to apply the calibration equation in the presence of vegetation. We cannot comment on whether the vegetation correction factor used here from [63] is descriptive of vegetation types beyond maize and soybean, especially those whose biomass water may be distributed differently. Proxies of BWE from remote sensing products (i.e., greenness, NDVI, and LAI) or crop growth calendars will likely provide reasonable estimates [74]. Note that the internal water content of row crop vegetation will vary greatly over the growing from approximately 90% to around 20–25% at the time of harvest. Again, crop calendars and crop growth stages will provide reasonable guesses.

### 4.5. Roadmap for Future Implementation

Some of the limitations in SWC measurement with gamma-ray spectroscopy may be overcome by following the trajectory of CRNS research over the past decade, which involves field validation, Monte Carlo simulation, and streamlining correction factors to develop a usable calibration function. Although the CRNS method detects particles with different production mechanisms and energy ranges, the attenuating power of hydrogen is the basis for SWC estimation with the CRNS and gSMS alike. One crucial step is to find methods to predict the unknown parameters in the calibration function and thereby reduce the number of calibration campaigns required. For instance, a multi-site study could explore whether there exists a relationship between $I_{0}$ and a or between the calibration parameters and any other known site-specific characteristics such as bulk density, soil type, and vegetation type or water content. The CRNS community has completed a diverse array of field studies to constrain the relationships between parameters and correction factors in the CRNS calibration equation [33,79] and the same commitment to a broad scope of field contexts is essential to future gSMS research.

The CRNS community has also taken advantage of simulation methods, such as Monte Carlo N-Particle Transport (MCNP) code and the Ultra Rapid Adaptable Neutron-Only Simulation (URANOS), to answer a variety of questions about footprint characteristics and sources of signal attenuation [73]. Topics explored in CRNS simulations that warrant study in gamma-ray simulations as well include footprint size for different energy ranges of detection [49,50,80], measurement capabilities in heterogenous landscapes [52,81], and performance in the presence of factors such as air humidity and fractional snow cover [82,83]. Corrections for water content in different types of vegetation such as agricultural row crops, forest, and grassland have also been advanced by Monte Carlo simulations [33,84,85].

Gamma-ray spectroscopy also has a history of using MCNP simulations to better understand the spectra generated by various gamma-ray source compositions and configurations [86,87,88,89,90] and has already begun to follow in the footsteps of CRNS research. For instance, ref. [63] used a Monte Carlo approach designed for assessing water content [54] to find the vegetation water correction factor used in this study. Similarly, ref. [53] used MCNP to predict gamma-ray spectra shape and intensity as a function of detector height for UAV applications, which are made feasible by the small payload of the gSMS compared to CRNS. Continuation of simulations and field studies can eventually enable accurate universal estimation of subfield-scale soil moisture through decreasing calibration labor and correcting for factors including vegetation water, detector height, heterogenous landscapes, and effective sensing depths.

## 5. Conclusions

Our analysis demonstrates the skill of the GRS method for estimating subfield SWC (a scale that describes a critical level of SWC variability). We found that the $I_{0}$ and a parameters must be fit to the data to accurately estimate our observed SWC data (RMSE = 0.033 g g^−1^) from ^40^K specific activity obtained from FSA. Biomass water was successfully accounted for by a correction factor that treats biomass as a layer of water over the soil surface. Further analysis to provide calibration recommendations found that the 27 observed SWC samples could be predicted with an RMSE of 0.035 g g^−1^ using a calibration procedure with only 10 profiles in the footprint and 5 calibrations. Future GRS research is tasked with reducing the number of calibrations required and continuing field validation and Monte Carlo simulation to actualize the full capabilities of GRS SWC estimation in a broad range of vegetation contexts and in both ground and UAV-based applications.

This work represents the most robust study to date bridging the divide between gamma-ray spectroscopy theory and field quantification of SWC. Up to this point, gamma-ray measurements have been primarily employed in research to detect relative change in SWC, specifically rain and irrigation events [91]. Quantification of SWC at the subfield scale is paramount for collecting data with the appropriate spatial variability for the major hydrological applications of our time, including flood forecasting, precision irrigation, and drought and wildfire monitoring. The validation, advancements, and opportunities for the gSMS method described in this study strongly support the future of hydrological and climate monitoring.

## Figures and Tables

**Figure 1 sensors-24-02223-f001:**
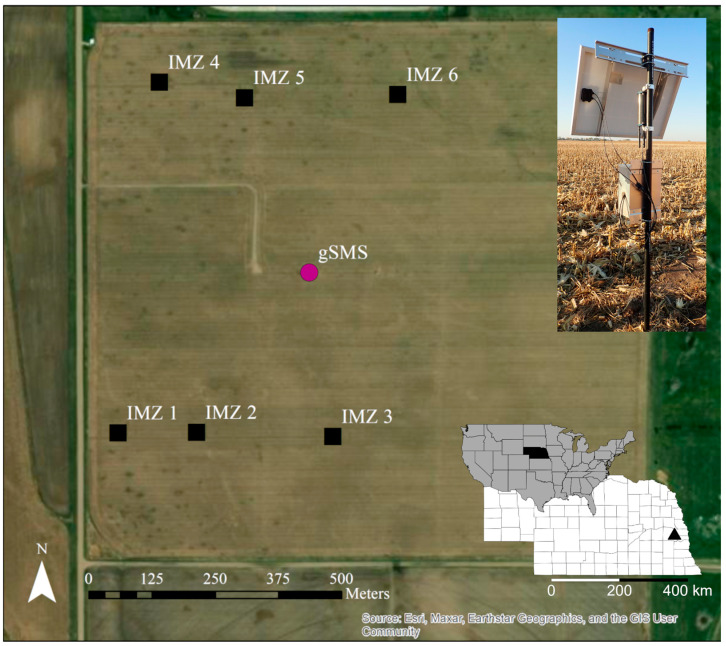
Locations of the gamma-ray soil moisture sensor (gSMS) and the intensive management zones (IMZs) in the field. A field photo of the installed gSMS is provided in the top right. The bottom right inset shows the central location of Nebraska within the United States and the location of the field (black triangle) in eastern Nebraska.

**Figure 2 sensors-24-02223-f002:**
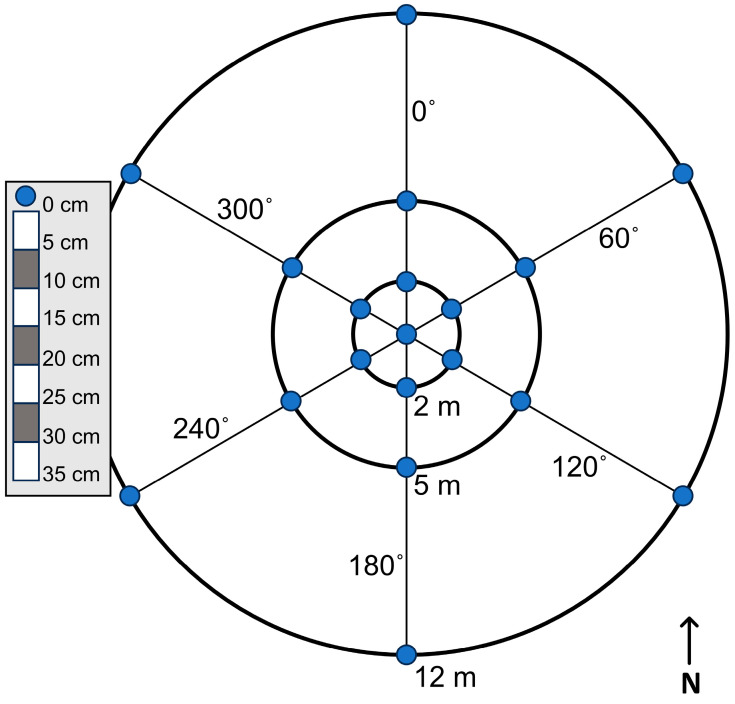
For each sampling campaign, samples were collected at 19 locations (blue points) surrounding the gSMS. At each of the 19 locations, 7 samples were collected over 5 cm depth intervals as depicted on the left.

**Figure 3 sensors-24-02223-f003:**
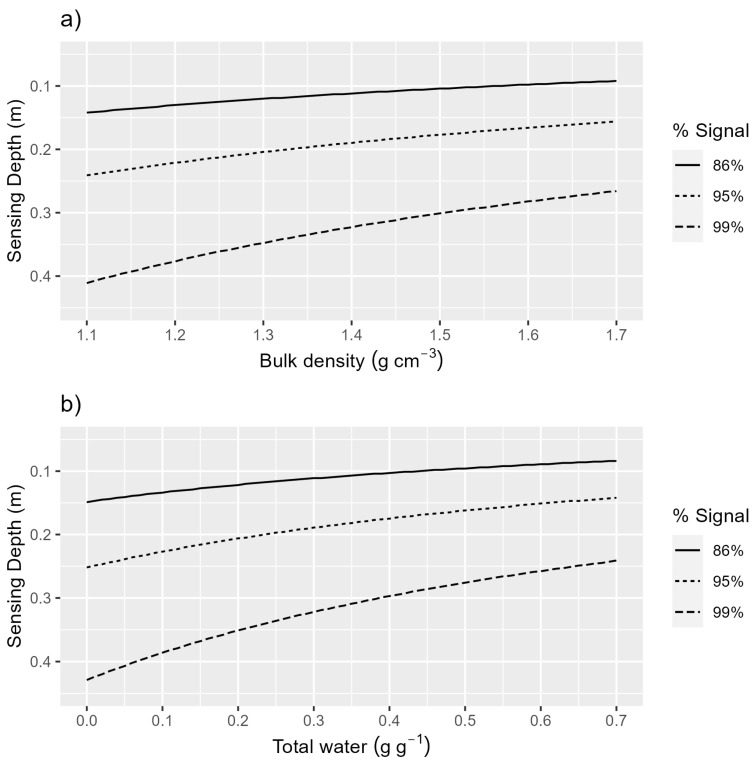
(**a**) Total water content is 0.27 g g^−1^ and the sensing depth for 86% (2 e-folding depth), 95% (3 e-folding depth), and 99% of the detected signal is plotted for a range of bulk densities using Equation (8). (**b**) Bulk density is 1.37 g cm^−3^ and the sensing depth for 86% (2 e-folding depth), 95% (3 e-folding depth), and 99% of the detected signal is plotted for a range of total water values using Equation (8).

**Figure 4 sensors-24-02223-f004:**
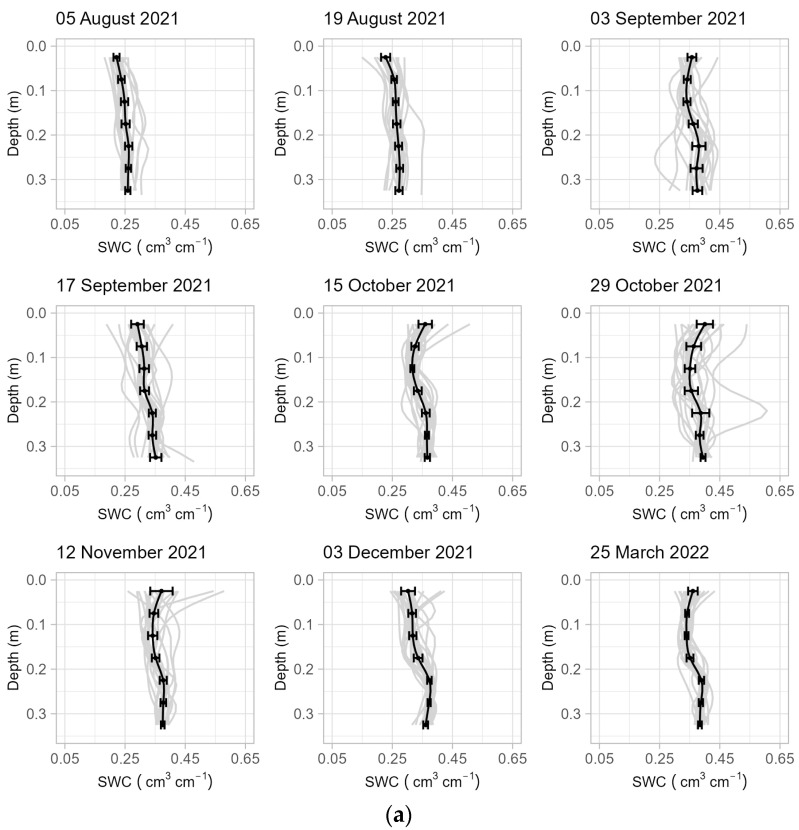
(**a**) Soil moisture profiles from the September 2021 to March 2022 sampling campaigns. The samples from the 7 depths at each location were interpolated to plot the 19 profiles for each sampling day in gray. For each sampling day, the samples were also averaged by depth (black points) and interpolated to find the average profile (black line), with horizontal error bars showing two standard errors. (**b**) Soil moisture profiles for the April to October 2022 sampling campaigns. (**c**) Soil moisture profiles for May to October 2023 sampling campaigns.

**Figure 5 sensors-24-02223-f005:**
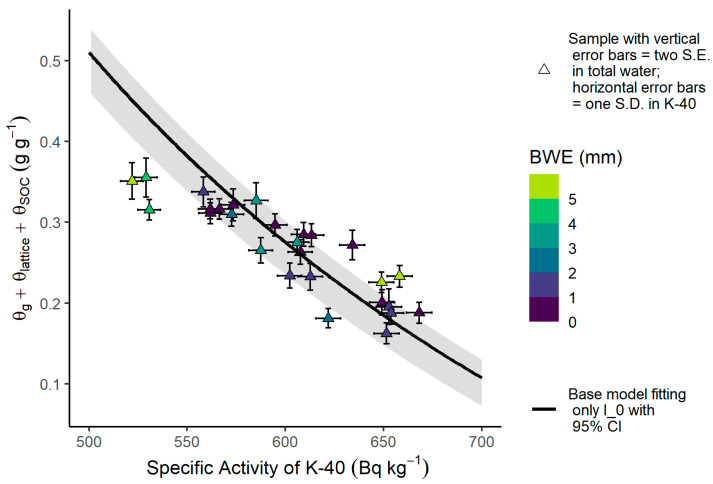
The experimental relationship between total water—the sum of gravimetric water content ($\theta_{g}$), lattice water ($\theta_{lattice})$, and soil organic carbon ($\theta_{SOC}$)—and ^40^K specific activity compared to the relationship predicted by the first calibration equation without the biomass correction (black line), with the corresponding 95% confidence interval (gray band).

**Figure 6 sensors-24-02223-f006:**
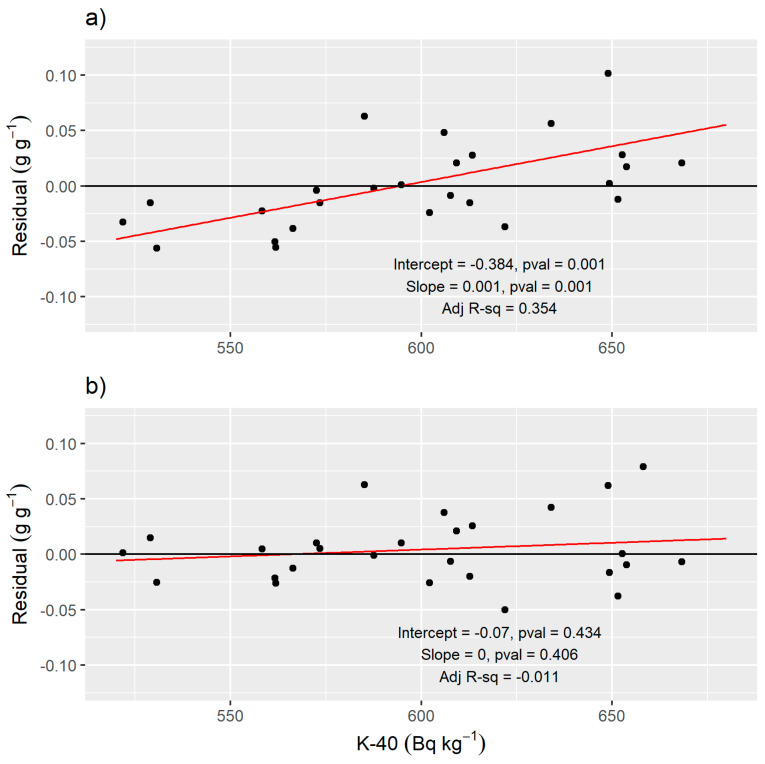
(**a**) Residuals from Equation (15). The red line shows the significant linear trend (slope and intercept *p*-values < 0.01). (**b**) Residuals from Equation (16) do not show a significant trend.

**Figure 7 sensors-24-02223-f007:**
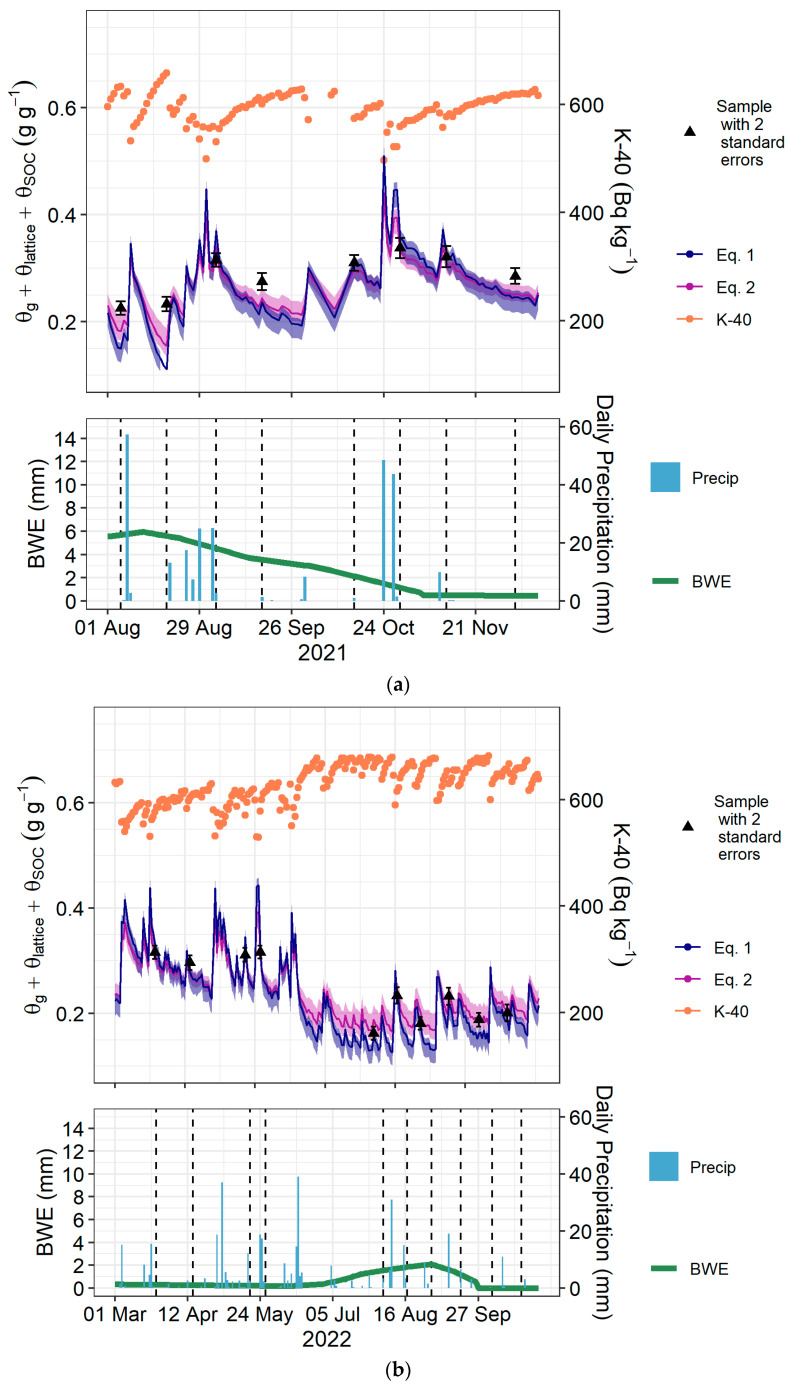
(**a**) A time series summary of the 2021 sampling period. In the upper panel are the daily ^40^K data, the gravimetric sample data, and the predictions from the first and second calibration equations (Equation (15) and Equation (16), respectively) with 95% confidence intervals calculated by bootstrapping. The lower panel shows precipitation events and estimated biomass water equivalence (BWE). Vertical dashed lines are the sampling dates. (**b**) A time series summary of the 2022 sampling period. (**c**) A time series summary of the 2023 sampling period.

**Figure 8 sensors-24-02223-f008:**
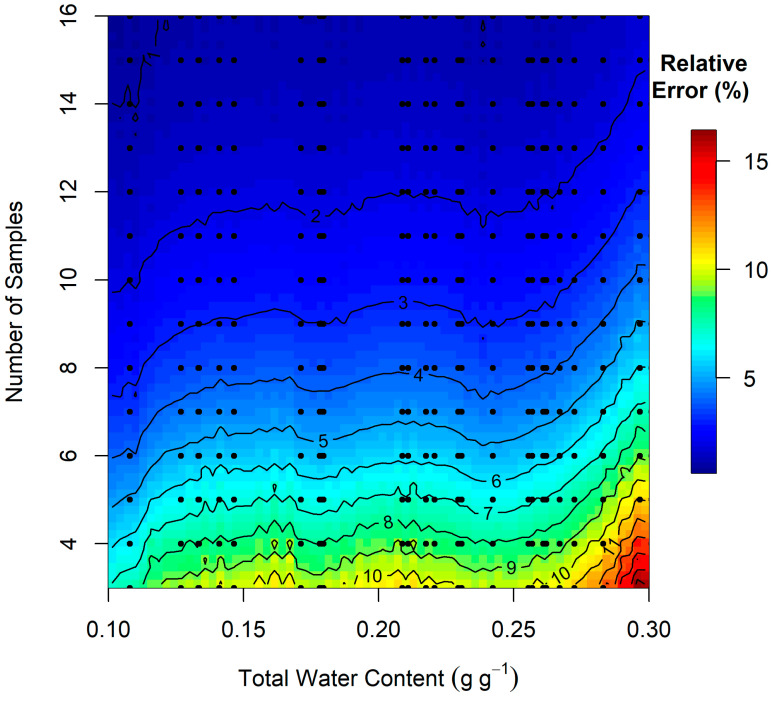
Relative error in total water content ($\theta_{tot}$) calculated from the number of sample profiles indicated on the vertical axis compared to $\theta_{tot}$ calculated using all 19 sample profiles. The image was generated by smoothing the sample relative error values shown by the black dots to a regularly spaced grid and then interpolating via inverse distance weighting.

**Figure 9 sensors-24-02223-f009:**
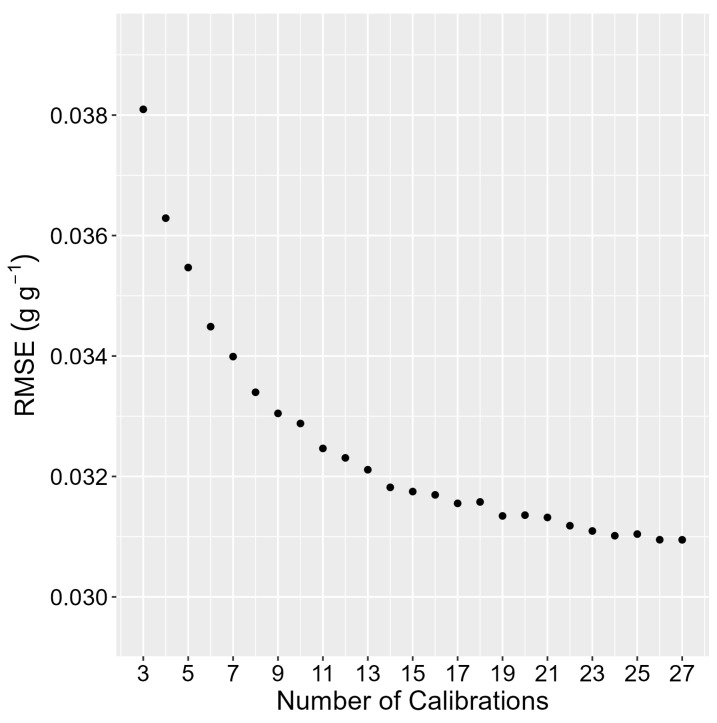
Root mean squared error (RMSE) in predicting total water content for all 27 samples, using Equation (16) calibrated with the number of calibrations on the horizontal axis and 10 randomly selected profiles from each selected sample day.

**Table 1 sensors-24-02223-t001:** Accuracies of major SWC sensing methods (from field validation studies when possible) are summarized using root mean square error (RMSE) or root mean square deviation (RMSD). Prominent advantages and disadvantages of each method are also provided.

Method	Accuracy	Advantages	Disadvantages	References
Neutron probe	<0.010 m^3^ m^−3^ RMSE	Not affected by temperature, high accuracy and sensitivity	Active radiation source, does not monitor continuously, low spatial representativeness	[18]
Dielectric probes	0.010 to 0.041 m^3^ m^−3^ RMSD	Can monitor continuously, commercially available and easy to operate	Low spatial representativeness, potential for installation error (air gaps), site-specific calibration required for best accuracy	[25,26]
Ground penetrating radar	0.030 m^3^ m^−3^ RMSE	Can map SWC at various spatial scales	Processing advancements cannot be applied by non-experts	[34,35]
Global positioning system	0.035 m^3^ m^−3^ RMSE	Represents 10 s to 100 s of meters, availability of GPS signals	Further research required to standardize technique, shallow measurement depth (~5 cm), vegetation introduces error	[23,36]
Cosmic-ray neutron sensor	0.010 to 0.040 m^3^ m^−3^ RMSE	Higher spatial representativeness, continuous monitoring	Isolating SWC signal from other hydrogen pools can be challenging, corrections require expertise	[37,38,39,40,41]
Soil Moisture Active Passive Mission (microwave remote sensing)	0.040 m^3^ m^−3^ (target error)	Higher spatial representativeness (~10 km)	Shallow measurement depth (~5 cm), low temporal resolution, vegetation introduces error	[42]

**Table 2 sensors-24-02223-t002:** Surface vegetation type is reported for the 27 gravimetric water content sample dates.

Sample Date	Vegetation	Sample Date	Vegetation
5 August 2021	Maize	31 August 2022	Soybean
19 August 2021	Maize	17 September 2022	Soybean
3 September 2021	Maize	5 October 2022	Bare soil
17 September 2021	Maize	22 October 2022	Bare soil
15 October 2021	Maize	15 May 2023	Maize
29 October 2021	Maize	8 June 2023	Maize
12 November 2021	Maize stover	21 June 2023	Maize
3 December 2021	Maize stover	10 July 2023	Maize
25 March 2022	Maize stover	24 July 2023	Maize
15 April 2022	Maize stover	9 August 2023	Maize
18 May 2022	Maize stover	28 August 2023	Maize
27 May 2022	Maize stover	21 September 2023	Maize
3 August 2022	Soybean	23 October 2023	Maize stover

**Table 3 sensors-24-02223-t003:** Weights calculated for use in the depth-weighted average (Equation (13)). The lower bound of the last weighting interval was set to 0.33 m; the theoretical depth from which 99% of the gamma-ray signal is expected to originate for a homogenous source with infinite radius and fixed depth with bulk density 1.37 g cm^−3^ and total water content of 0.27 g g^−1^.

Soil Sample Depth Interval (m)	Weight
0–0.05	0.54
0.05–0.1	0.23
0.10–0.15	0.11
0.15–0.20	0.059
0.20–0.25	0.032
0.25–0.30	0.018

**Table 4 sensors-24-02223-t004:** Site-specific lattice water and soil organic carbon values for the USNe-3 field from 2023. Values are depth-weighted with weights found using the same method applied in Table 2.

Depth (cm)	Lattice Water (g g^−1^)	Soil Organic Carbon Water (g g^−1^)
0–10	0.049	0.005
10–20	0.049	0.004
20–30	0.054	0.004
Weighted	0.049	0.005

**Table 5 sensors-24-02223-t005:** Site-specific bulk density values from for the USNe-3 field from 2023. Values are depth-weighted with weights found using the same method applied in Table 3. NA values are inserted where one or fewer bulk density samples were obtained for a given depth. Uncertainty values are the standard error.

Depth (cm)	Bulk Density (g cm^−3^)			
	8 June 2023	9 August 2023	23 October 2023	Average
0–5	NA	NA	1.18 ± 0.09	1.18 ± 0.09
5–10	NA	1.04 ± 0.08	1.42 ± 0.06	1.23 ± 0.07
10–15	1.36 ± 0.09	1.37 ± 0.07	1.52 ± 0.06	1.42 ± 0.07
15–20	1.44 ± 0.05	1.52 ± 0.05	1.49 ± 0.05	1.48 ± 0.05
20–25	1.46 ± 0.03	1.5 ± 0.09	1.41 ± 0.05	1.46 ± 0.06
25–30	1.37 ± 0.05	1.46 ± 0.05	1.51 ± 0.05	1.45 ± 0.05
			Weighted:	1.26 ± 0.08

**Table 6 sensors-24-02223-t006:** Validation statistics and parameter values after fitting the first calibration equation (Equation (15)) and the second calibration equation (Equation (16)) with the 27 samples using the shuffled complex evolution algorithm (sceua function in the R package, rtop v0.6.6). The $I_{0}$ and *a* parameters were fit to the data (where applicable), and root mean squared error (RMSE) and R^2^ values were calculated using leave-one-out cross-validation. At the ^40^K peak energy of 1.46 MeV, the value for ${\left(\mu /\rho \right)}_{s}$ = 0.05257 cm^2^ g^−1^ and the value ${\left(\mu /\rho \right)}_{w}$ = 0.05836 cm^2^ g^−1^.

Equation	RMSE (g g^−1^)	R^2^	Adj R^2^	$I_{0}$(Bq kg^−1^)	*a*
15	0.045	0.34	0.25	792	NA
16	0.033	0.66	0.59	897	0.63

## Data Availability

The processed data presented in this study are openly available in GitHub (Appendix A).

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
