# Peer review of "Field Testing of Gamma-Spectroscopy Method for Soil Water Content Estimation in an Agricultural Field"

_sensors, 2024, doi:10.3390/s24072223_

Round 1
Reviewer 1 Report
Comments and Suggestions for Authors
The manuscript presents a thorough investigation into the application of GRS for estimating SWC at a subfield scale. The authors conducted a three-year field validation study, which involved extensive gravimetric water content sampling campaigns and measurements from a stationary GRS sensor in maize and soybean fields in Nebraska, United States. While the study addresses an important gap in the accurate estimation of SWC from GRS in the field, there are several concerns. Firstly, the manuscript primarily focuses on verifying the performance of the sensor and calibrating it for SWC estimation. While this is a crucial aspect, it is not sufficient for publication in this journal. The paper should strive to provide novel insights or advancements beyond mere sensor validation and calibration. Secondly, although the authors propose adjustments to the theoretical equations to better describe the field data, the significance of these adjustments and their implications for broader hydrological applications need to be more thoroughly discussed. The study falls short of meeting the standards for publication in this journal.
Reviewer 2 Report
Comments and Suggestions for Authors
Comments for authors are in the attached file.

Reviewer 3 Report
Comments and Suggestions for Authors
The authors claim that using a CsI (Cesium-Iodide) gSMS (gamma Spectroscopy Measurement System) to measure soil water content in the crop field could be a new and innovative technique. In fact, this study follows a trajectory of research and application similar to that of Cosmic-Ray Neutron Sensing (CRNS) over the past decade. This trajectory includes field validation, Monte Carlo simulation, and the streamlining of correction factors to develop a usable calibration function. This indicates that while the specific application of CsI gSMS for SWC measurement in agricultural fields may incorporate novel elements or specific innovations, the broader methodological framework—particularly the integration of gamma-ray spectroscopy with field validation and simulation techniques for SWC estimation—aligns with established practices in the field. The authors did not explicitly state that this application is the first of its kind. However, it implies an ongoing development and refinement process that leverages new and existing technological and methodological applied to a critical field for humankind's survival - Agriculture.
In the presented work, a 100 ml CsI gSMS developed by Medusa Radiometrics was employed to detect gamma rays, which originated naturally from potassium-40 present in the soil. The primary objective of this research was to conduct a field test of a gamma-spectroscopy method aimed at estimating soil water content within an agricultural context. This effort sought to effectively bridge the theoretical and experimental data divide, enabling accurate field estimation of soil water content utilizing gamma-ray spectroscopy (GRS).
The study's results demonstrated the CsI gSMS's capability in capturing a variety of heterogeneous soil moisture profiles, with observations of wetting fronts in specific instances (e.g., on 10 July 2023 and 24 July 2023). The relationship between observed soil moisture and naturally occurring potassium-40 generally aligned with what the authors expected to read. However, deviations were noted in both the dry and wet extremes of the soil water content (SWC) range. The authors decided to apply a biomass correction factor, notably improving the root mean squared error (RMSE) from 0.052 g/g to 0.045 g/g and the adjusted R^2 from 0.05 to 0.25, illustrating the correction's effectiveness in enhancing model accuracy. Further refinement using a second calibration equation addressed trends in the residuals, leading to improved goodness of fit statistics, including RMSE and adjusted R^2 values, compared to the initial calibration equation.
The authors acknowledge that overcoming the limitations in SWC measurement with gamma-ray spectroscopy and enhancing its application will require a concerted effort akin to the developments seen with CRNS research over the last decade. This approach encompasses extensive field validation, the application of Monte Carlo simulations, and the refinement of correction factors to devise a practical calibration function. Despite the differences in detection mechanisms and energy ranges between the CRNS method and the gSMS, both rely on the attenuation properties of hydrogen for estimating SWC. A key challenge for future implementation is developing methodologies to predict the calibration function's unknown parameters, thus reducing the need for extensive calibration campaigns. This commitment to exploring a wide range of field conditions is crucial for the future success and reliability of gSMS in SWC estimation.
The authors have employed rigorous methodologies in demonstrating that soil water content (SWC) estimation through gamma Spectroscopy Measurement System (gSMS) presents a viable alternative to current techniques, such as the traditional gravimetric sampling, striving to position gSMS as a novel approach in the field. They have also been transparent about the obstacles in making this sensing method more reliable and universally applicable, acknowledging the need for further research and calibration. Given their thorough approach and honesty about the challenges faced, this work is of significant merit and warrants publication in the respected journal Sensors (MDPI), as it contributes valuable results and opens new strategies for advancements in SWC estimation techniques.
While this work is undoubtedly valid for publication in Sensors due to its rigorous methodology and transparent discussion of challenges, it is necessary to improve the presentation quality of Figures 4, 5, and 6. The current versions appear to have been directly copied and pasted, resulting in low-resolution images that detract from the overall clarity and professionalism of the manuscript. Enhancing the resolution and visual quality of these figures will significantly contribute to the readers' understanding and appreciation of the research findings.
Comments on the Quality of English LanguageMinor editing of English language required
Round 2
Reviewer 1 Report
Comments and Suggestions for Authors
I will remain consistent with my previous comments.
The authors must include all recent articles, preferably review articles, discussing different soil moisture sensors for in-situ measurement of soil water content, for critical comparison.
Reviewer 2 Report
Comments and Suggestions for Authors
All the observations and findings made were taken into consideration and adequately modified in revision 2 of the paper, therefore I suggest the acceptation.
Author Response
Thank you for your time and positive feedback on this manuscript. A revised manuscript is provided with tracked changes in response to another reviewer.